# PGC7 Regulates Genome-Wide DNA Methylation by Regulating ERK-Mediated Subcellular Localization of DNMT1

**DOI:** 10.3390/ijms24043093

**Published:** 2023-02-04

**Authors:** Xing Wei, Yingxiang Liu, Weijie Hao, Peiwen Feng, Lei Zhang, Hongni Xue, Qunli Zhou, Zekun Guo

**Affiliations:** 1College of Veterinary Medicine, Northwest A&F University, Yangling, Xianyang 712100, China; 2Key Laboratory of Animal Biotechnology, Ministry of Agriculture and Rural Affairs, Northwest A&F University, Yangling, Xianyang 712100, China; 3College of Life Science, Northwest A&F University, Yangling, Xianyang 712100, China

**Keywords:** PGC7, DNA methylation, ERK, phosphorylation, DNMT1

## Abstract

DNA methylation is an epigenetic modification that plays a vital role in a variety of biological processes, including the regulation of gene expression, cell differentiation, early embryonic development, genomic imprinting, and X chromosome inactivation. PGC7 is a maternal factor that maintains DNA methylation during early embryonic development. One mechanism of action has been identified by analyzing the interactions between PGC7 and UHRF1, H3K9 me2, or TET2/TET3, which reveals how PGC7 regulates DNA methylation in oocytes or fertilized embryos. However, the mechanism by which PGC7 regulates the post-translational modification of methylation-related enzymes remains to be elucidated. This study focused on F9 cells (embryonic cancer cells), which display high levels of PGC7 expression. We found that both knockdown of *Pgc7* and inhibition of ERK activity resulted in increased genome-wide DNA methylation levels. Mechanistic experiments confirmed that inhibition of ERK activity led to the accumulation of DNMT1 in the nucleus, ERK phosphorylated DNMT1 at ser717, and DNMT1 Ser717-Ala mutation promoted the nuclear localization of DNMT1. Moreover, knockdown of *Pgc7* also caused downregulation of ERK phosphorylation and promoted the accumulation of DNMT1 in the nucleus. In conclusion, we reveal a new mechanism by which PGC7 regulates genome-wide DNA methylation via phosphorylation of DNMT1 at ser717 by ERK. These findings may provide new insights into treatments for DNA methylation-related diseases.

## 1. Introduction

DNA methylation is an epigenetic modification in which DNA methyltransferase adds a methyl to the fifth position of the cytosine in cytosine-phosphate-guanine (CpG) motifs and non CpG dinucleotides [1,2]. DNA methylation plays an essential role in regulating gene expression, cell differentiation, early embryonic development, genomic imprinting, and X chromosome inactivation [3]. The DNA methylation pattern of the mammalian genome forms during gametogenesis and early embryonic development, and is established and maintained by the de novo DNA methyltransferases DNMT3a and DNMT3b and the maintenance methyltransferase DNMT1. All three of these are members of the DNA methyltransferase (DNMT) family [4,5]. Although DNMT3L lacks enzyme activity, it has two functions: binding to the unmethylated histone H3K4 and activating DNMT3a, which is necessary for the establishment of maternal methylation imprinting [6,7]. Abnormal DNA methylation is associated with several human diseases, including neurodegenerative, neurological, and autoimmune diseases. In addition, two patterns of abnormal methylation are characteristic of cancer cells: hypermethylation of specific CpG islands and/or their shores and genome-wide hypomethylation [8,9,10].

PGC7 (also known as Stella or Dppa3) is a maternal effector protein that plays an indispensable role in primordial germ cells, oocytes, and preimplantation embryos, since it is involved in the maintenance of imprinted gene methylation during early embryonic development [11,12,13]. In oocytes, PGC7 affects the localization of DNMT1 by regulating the subcellular localization of UHRF1, thereby preventing high levels of DNMT1 in the nucleus and thus maintaining the stability of the oocyte genome [14]. In fertilized embryos, PGC7 binds to H3K9 me2 to inhibit the active demethylation of female prokaryotic imprinted genes by TET3. It also maintains DNA methylation stability of imprinted genes in both embryonic [15] and somatic cells. In HEK293T cells, PGC7 has been found to interact with TET2 and TET3, inhibiting their enzyme activity and protecting CpG methylation of imprinting sites from TET-mediated DNA demethylation [16].

In somatic cells, PGC7 leads to reduced genome-wide DNA methylation levels via different pathways. In PGC7-expressing NIH3T3 cells, PGC7 was found to disrupt the colocalization of UHRF1 and DNMT1 with proliferating cell nuclear antigen. It did so by directly binding to UHRF1, thereby preventing DNMT1 from maintaining DNA methylation [17]. In another study, in which HEK293T cells overexpressing PGC7 or a PGC7 mutant lacked nuclear export activity (i.e., L44A/L46A), PGC7 was found to bind to the PHD domain of UHRF1, causing it to unbind from the unmethylated histone H3K9 or its methylated counterpart H3K9 me3. The effect of this was to destroy the binding of UHRF1 to chromatin, thereby reducing the affinity of DNMT1 towards chromatin and consequently decreasing DNA methylation levels [18]. However, whether PGC7 regulates the function of DNMT1 has not been reported.

Post-translational modifications of DNMT1—especially phosphorylation—also affect the localization, stability, or activity of DNMT1 and thus regulate DNA methylation. Two sites within the DNMT1 nuclear localization signal (NLS) can be phosphorylated by AKT kinase. This can be activated by interleukin-6 (IL-6), which promotes the nuclear translocation of DNMT1 and thus leads to changes in DNA methylation [19]. In addition, the phosphorylation of DNMT1S143 by AKT1 maintains DNMT1 stability by inhibiting the methylation of DNMT1K142 by SET7, which generally reduces DNMT1 protein stability [20]. In addition to localization and stability, phosphorylation modifications also affect DNMT1 activity. For example, the phosphorylation of DNMT1 ser515 enables its methyltransferase activity [21]. In addition, it has been reported that MEK/ERK phosphorylation regulates the expression or activity of DNMT1, thereby causing DNA methylation changes that affect the expression of tumor suppressor genes in prostate cancer as well as methylation-sensitive genes associated with the pathogenesis of systemic lupus erythematosus (SLE). Histone deacetylase inhibitors inhibit the growth of prostate cancer cells by inhibiting ERK activity and downregulating DNMT1 protein levels, and were found to reverse promoter hypermethylation and gene silencing of three tumor suppressor genes in an LNCaP prostate cancer cell line; these three genes included retinoic acid receptor B2 (RARB2) and cycle-dependent kinase inhibitors p21 and p16 [22]. Finally, another study of T cells sourced from patients with SLE showed that protein phosphatase 2A catalytic subunit (PP2Ac) downregulates DNMT1 expression and attenuates DNMT1 enzyme activity by inhibiting MEK/ERK phosphorylation. This in turn reduced DNA methylation levels and resulted in increased expression of methylation-sensitive genes related to SLE pathogenesis and increased SLE pathophysiology [23]. However, to date, it remains unclear whether ERK directly phosphorylates DNMT1 and regulates DNA methylation.

In this study, we show that in F9 cells that show high levels of PGC7 expression, PGC7 affects the localization of DNMT1 and thus regulates genome-wide DNA methylation via an ERK pathway-mediated DNMT1 phosphorylation modification at ser717. Moreover, we propose a new mechanism by which PGC7 can regulate genome-wide DNA methylation through this pathway.

## 2. Results

### 2.1. PGC7 Regulates Genome-Wide DNA Methylation in F9 and NIH3T3 Cells

Previous studies have reported that PGC7 can regulate DNA methylation [14,15,16,17,18]. We first evaluated the effects of PGC7 by knocking down *Pgc7* in F9 cells and used Si*Pgc7*-FAM, a green fluorescent dye marker, to determine genome-wide 5 mC levels following methylation staining. Our results showed that knocking down *Pgc7* increased genome-wide 5 mC levels (Figure 1a,b). Dot blot results confirmed that genome-wide 5 mC levels had increased after knocking down *Pgc7* (Figure 1c). We also overexpressed PGC7 in both F9 and NIH3T3 cell lines and assessed genome-wide 5 mC levels by a dot blot assay. These results showed that overexpression of PGC7 reduced genome-wide 5 mC levels (Figure 1d,e). In addition, methylation staining confirmed that overexpression of PGC7 in HEK293T cells reduced genome-wide 5 mC levels (Appendix A). Taken together, these results suggest that PGC7 regulates genome-wide DNA methylation.

### 2.2. Inhibition of ERK Activity Improves Genome-Wide DNA Methylation

It has been reported that the ERK pathway is involved in the regulation of DNA methylation by histone deacetylase inhibitors and PP2Ac [22,23]. To confirm this, we treated F9 cells with the MEK/ERK inhibitor PD0325901 (1 μM, 12 h) (hereafter referred to as PD) and assessed genome-wide 5 mC levels by methylation staining. Our results showed that inhibition of ERK activity increased genome-wide 5 mC levels (Figure 2a,b). We also used a dot blot test to assess genome-wide 5 mC levels of F9 and NIH3T3 cells treated with PD, and confirmed that PD treatment increased genome-wide 5 mC levels (Figure 2c,d). These results suggested that ERK activity also regulates genome-wide DNA methylation.

The increase in DNA methylation in response to treatment with PD may be caused by induced differences in the expression or activity of DNMT1. To further study whether ERK activity affects the expression of DNMT1, we treated both F9 and NIH3T3 cells with PD and assessed DNMT1 protein expression by Western blot. We found that PD treatment resulted in reduced DNMT1 protein expression (Figure 2e). In addition, we also found that genome-wide 5 mC levels increased following PD treatment, which suggests that ERK may regulate DNMT1 activity to increase genome-wide DNA methylation.

We then hypothesized that changes in activity probably reflect changes in DNMT1 localization, and therefore examined the subcellular localization of endogenous DNMT1 protein after PD treatment by immunostaining. We found that PD treatment promoted the nuclear localization of DNMT1 (Figure 2f,g), suggesting that increased DNA methylation following PD treatment is likely due to altered DNMT1 localization. Taken together, these results suggest that ERK may affect genome-wide DNA methylation by regulating the subcellular localization of DNMT1.

### 2.3. PGC7 Regulates ERK Activity by Regulating Mek1 Expression

It has been reported that PGC7 binds to UHRF1, thereby disrupting the localization of DNMT1 and resulting in genome-wide DNA hypomethylation in PGC7-expressing NIH3T3 cells [17]. In oocytes, PGC7 regulates the subcellular localization of UHRF1, thus affecting the localization of DNMT1 and preventing high levels of DNMT1 from entering the nucleus; this maintains the stability of oocyte genome [14]. Given these findings, we speculate that PGC7 may regulate DNA methylation by interacting with ERK or by regulating ERK activity. To test this hypothesis, we knocked down *Pgc7* in F9 cells and assessed the phosphorylation levels of ERK by Western blot. These results showed that knocking down *Pgc7* downregulated ERK phosphorylation (Figure 3a). To further explore how PGC7 may regulate ERK phosphorylation, we knocked down *Pgc7* in F9 cells and determined the effect of PGC7 on the transcription levels of ERK1/2 and its upstream kinase MEK1/2 via RT-qPCR. We also found that knockdown of *Pgc7* significantly downregulated *Mek1* transcription but did not affect *Erk1/2* transcription (Figure 3b); this result suggests that PGC7 may affect ERK phosphorylation by regulating *Mek1* transcription. At the same time, we also explored the possible regulation of ERK phosphorylation by PGC7 by analyzing their interactions. Co-IP showed that PGC7 did not interact with ERK1/2 (Appendix A), indicating that the regulation of ERK phosphorylation by PGC7 was not realized via direct interaction between PGC7 and ERK1/2. Taken together, these results suggest that PGC7 may regulate ERK phosphorylation by regulating *Mek1* transcription.

### 2.4. ERK Phosphorylates DNMT1 at Ser717 and Regulates the Subcellular Localization of DNMT1

The inhibition of ERK activity promotes the nuclear localization of DNMT1, and we speculate that DNMT1 may act as a substrate of ERK, and that its localization is regulated by ERK phosphorylation. To verify this hypothesis, we first verified the interaction between ERK1 and DNMT1 by co-IP (Figure 4a,b). By consulting the Scansite and PhosphoSitePlus phosphorylation mass spectrometry databases, we found that there are many phosphorylation modification sites on DNMT1, but little is known about the kinases that target these sites for phosphorylation modification or about the effect that these phosphorylation modifications have on DNMT1 function. The potential ERK phosphorylation sites on DNMT1 were further predicted by GPS5.0, which resulted in the identification of three high-confidence ERK1/2 phosphorylation sites: ser717 (S717), ser958 (S958), and ser1421 (S958) (Figure 4c). We then determined the degree of evolutionary conservation in the DNMT1 protein sequences of different species, finding that sequences near these three sites were highly conserved, with the sequence near S717 showing the highest conservation (Figure 4d). This finding was consistent with the sequence preference, i.e., Pro-xxx-Ser/Thr-Pro, of ERK phosphorylation of substrates [24].

To explore whether these three phosphorylation sites affect the localization of DNMT1 as well as whether they are affected by ERK activity, we constructed mutant expression vectors in which these three serine residues were mutated into alanine. These vectors were then overexpressed in NIH3T3 cells so that we could then examine the effect of PD treatment on the localization of the three DNMT1 mutants relative to wild-type DNMT1. Our results showed that the subcellular localization of wild-type DNMT1 was consistent with the localization and distribution of endogenous DNMT1, in that it was mainly distributed in the nucleus but was also present to a lesser degree in the cytoplasm. Compared to DMSO treatment, PD treatment also promoted the accumulation of wild-type DNMT1 in the nucleus (Figure 4e). In the control group, the subcellular localization of DNMT1S958A and DNMT1S1421A mutants was distributed mainly in the nucleus and to a lesser extent in the cytoplasm, which was the same as in the wild type. PD treatment also promoted the nuclear localization of DNMT1S958A and DNMT1S1421A, confirming that these two sites are not regulated by ERK activity. In the control group, the subcellular localization of the DNMT1S717A mutant protein was in the nucleus, and this was consistent with the localization of wild-type DNMT1 following PD treatment. After PD treatment, the localization of the DNMT1S717A mutant protein in the nucleus did not change (Figure 4e), indicating that S717 is likely to be a phosphorylation target of ERK, and that phosphorylation of this site affects the subcellular localization of DNMT1.

Next, we examined the subcellular localization of wild-type DNMT1 and the DNMT1S717A mutant protein in NIH3T3 and HEK293T cells after PD treatment. We found that wild-type DNMT1 was mainly localized in the nucleus, with some present in the cytoplasm, and showed nuclear localization following PD treatment. The localization pattern of the DNMT1S717A mutant protein was also primarily nuclear, and did not change following PD treatment (Figure 4f). To further confirm that S717 was the ERK phosphorylation site, we created two truncated wild-type DNMT1 and DNMT1S717A mutants (i.e., DNMT1△WT and DNMT1△S717A) and constructed expression vectors including the truncated proteins (Figure 4g). We then overexpressed wild-type DNMT1, the DNMT1S717A mutant, or one of the truncated DNMT1△WT and DNMT1△S717A vectors in HEK293T cells. After PD treatment, protein immunoprecipitation was performed to determine the effect of ERK on overall serine phosphorylation. We found that PD treatment inhibited overall serine phosphorylation of DNMT1 to a greater degree than DMSO treatment in the group overexpressing wild-type DNMT1 (Figure 4h, lane 3 versus lane 2). Moreover, the overall serine phosphorylation level of DNMT1S717A was also significantly lower than wild-type DNMT1 (Figure 4h, lane 4 versus lane 2). We also found that the overexpressed DNMT1S717A mutant group showed serine phosphorylation levels of DNMT1S717A that did not change significantly following PD treatment when compared to DMSO treatment (Figure 4h, lane 5 versus lane 4). Furthermore, an immunoprecipitation assay of DNMT1△WT and DNMT1△S717A confirmed that PD treatment reduced overall serine phosphorylation levels relative to DMSO treatment in the overexpressed DNMT1△WT group (Figure 4i, lane 3 versus lane 2). The overall serine phosphorylation level of DNMT1△S717A was also reduced compared to the truncated wild type (Figure 4i, lane 4 versus lane 2). In addition, in the group overexpressing DNMT1△S717A, the overall serine phosphorylation level of DNMT1△S717A did not change significantly following PD treatment compared with DMSO (Figure 4i, lane 5 versus lane 4). Taken together, these results are consistent with the hypothesis that S717 is the phosphorylation site of ERK.

PD treatment was found to lead to the accumulation of DNMT1 in the nucleus, which was coincident with increases in genome-wide DNA methylation levels. Therefore, we speculate that the nuclear translocation of DNMT1 is one cause of the increased genome-wide DNA methylation. The DNMT1S717A mutant was also mainly located in the nucleus, and this was consistent with the localization of DNMT1 after PD treatment. To verify the effect of nuclear translocation of DNMT1 on genome-wide DNA methylation levels, we overexpressed wild-type DNMT1 and DNMT1S717A mutants in HEK293T cells, extracted genomic DNA, and performed a dot blot assay to determine the effect of the DNMT1S717A mutant on genome-wide DNA methylation. These results showed that the DNMT1S717A mutant increased genome-wide 5 mC levels to a greater degree than the wild type (Figure 4j).

### 2.5. PGC7 Regulates the Subcellular Localization of DNMT1 through ERK

Figure 2f,g and Figure 3a show that PGC7 downregulated ERK phosphorylation, and the inhibition of ERK activity led to the accumulation of DNMT1 in the nucleus. Therefore, we speculate that PGC7 may regulate DNMT1 localization by regulating ERK phosphorylation. To test this hypothesis, *Pgc7* was knocked down in F9 cells, and we assessed the subcellular localization of endogenous DNMT1 via immunofluorescence. We found that knockdown of *Pgc7* promoted the nuclear localization of DNMT1 (Figure 5a) and resulted in decreased ERK activity (Figure 5b,c). In contrast, overexpression of PGC7 in NIH3T3 cells weakened the nuclear localization of endogenous DNMT1 (Appendix A). In PGC7-overexpressing HEK293T cells and *Pgc7*-knockout oocytes, the presence of PGC7 alters DNMT1 localization by affecting UHRF1 localization, and ultimately changes patterns of DNA methylation [14,18]. Therefore, we speculate that the changes in DNMT1 localization and genome-wide DNA methylation caused by knocking down *Pgc7* may have been caused by changes in the localization of UHRF1. However, our data indicate that UHRF1 localization did not change significantly after knocking down *Pgc7* (Figure 5d), which suggests that PGC7 regulates the subcellular localization of DNMT1 by regulating ERK phosphorylation, but not by regulating the localization of UHRF1.

## 3. Discussion

The maternal factor PGC7 has been reported to maintain DNA methylation during early embryonic development [12,25,26]. In PGC7-deficient primordial germ cells (PGCs), retrotransposon genes such as long interspersed nuclear element-1 and intracisternal A particle show increased 5 mC and decreased 5 hmC levels, indicating that PGC7 is involved in Tet-mediated active demethylation processes in PGC reprogramming [27]. In addition, forced expression of PGC7 has been found to induce overall DNA hypomethylation in NIH3T3 cells [28]. The effect of PGC7 on DNA methylation involves two distinct processes, i.e., active demethylation mediated by Tet and passive demethylation mediated by DNA methylation-related enzymes. In this study, we detected an increase in overall 5 mC levels in *Pgc7*-knockdown F9 cells and a decrease in F9 and NIH3T3 cells overexpressing PGC7. Taken together, these results reconfirmed the effect of PGC7 on DNA methylation. Previous studies have identified the mechanism by which PGC7 regulates DNA methylation as resulting from an interaction between PGC7 and H3K9 me2, TET2/TET3, or UHRF1. However, to date, the mechanism by which PGC7 acts on other levels, such as on post-translational modifications, remains unclear. Previous studies have found that the overall methylation loss of ES cells cultured with 2i (i.e., inhibitors of Mek1/2 and Gsk3β), including imprinting in regulatory regions, was mainly caused by inhibition of MEK1/2. Moreover, prolonged culture time was found to severely reduce the developmental potential of cells [29,30]. However, in this study, we found that the genome-wide 5 mC levels of F9 and NIH3T3 cells were increased after treatment with PD0325901, indicating that ERK may exert different effects in different cell lines. Moreover, knockdown of *Pgc7* and inhibition of ERK activity also resulted in increased genome-wide DNA methylation levels. This finding suggested that both PGC7 and ERK are involved in the regulation of DNA methylation, and that ERK is likely related to the regulation of DNA methylation by PGC7. We then determined that knockdown of *Pgc7* in F9 cells was associated with downregulation of ERK phosphorylation, so PGC7 likely participates in the regulation of DNA methylation by maintaining ERK activity. The regulation of DNA methylation is mainly achieved by altering the expression or activity of DNA methylation-related enzymes, and it is known that ERK regulates the expression and activity of these enzymes. In addition, Sunahori et al. found that the MEK/ERK pathway is involved in the regulation of DNMT1 expression and activity by PP2Ac in T cells of patients with SLE. They also found that this pathway affects the expression of methylation-sensitive genes involved in SLE pathogenesis and pathophysiology [23]. Sarkar et al. also found that ERK activity is involved in the regulation of DNMT1 expression by histone deacetylase inhibitors. This was found to affect the promoter methylation and gene expression of three tumor suppressor genes (i.e., RARB2, p21, and p16) which thereby affected the growth of prostate cancer cells [22]. Taken together, these studies suggest that ERK activity is directly involved in the regulation of DNA methylation by regulating the expression or activity of DNMT1. Zhang et al. found that the MEK/ERK signaling pathway is involved in cytoplasmic translocation of DNMT3a induced by high glucose, which led to the hypomethylation of connective tissue growth factors in human glomerular mesangial cells (hMSCs) [31]. We therefore speculated that ERK likely regulates the localization of DNMT1. In this study, we found that inhibition of ERK activity promoted nuclear subcellular localization of DNMT1, and this change in localization further promoted genome-wide DNA methylation. Therefore, the new pathway causing increased methylation levels in response to PGC7 knockdown likely affects the subcellular localization of DNMT1 via ERK activity. However, we then faced another question: how does ERK affect the subcellular localization of DNMT1?

Protein interactions are known to regulate the activity of DNMT1. Both the SET and ring finger-associated domains as well as the N-terminal ubiquitin-like domain (UBL) of UHRF1 directly interact with DNMT1. These interactions stimulate DNMT1 activity and increase its methylation activity on hemi-methylated DNA [32,33,34]. Another study found that IL-6 promotes the nuclear translocation of DNMT1 via phosphorylation of the NLS of DNMT1 by AKT kinase [19]. Moreover, it is also known that the phosphorylation of DNMT1 ser515 activates its methyltransferase activity [21]. The activity of DNMT3a is downregulated by CK2, which phosphorylates two key residues (i.e., S386 and S389) near the Pro-Trp-Trp-Pro domain of DNMT3a [35]. Moreover, fibroblast growth factor blocks the recruitment of DNMT3a to the promoter of the prochondrogenic transcription factor Sox9 by inducing phosphorylation of DNMT3a by ERK1/ERK2. This regulates Sox9 expression and ultimately controls cartilage formation in limb bud mesenchymal cells [36]. These studies demonstrate that the localization and activity of DNMT1/DNMT3a can be regulated by phosphorylation modifications. Therefore, we speculated that ERK likely affects the intracellular localization of DNMT1 via phosphorylation modification. In this study, we confirmed that S717 is an important site for ERK phosphorylation on DNMT1 using a site-directed mutagenesis strategy coupled with immunofluorescence and IP experiments. Marin et al. found that AMPK phosphorylates human DNMT1 at S730 (S717 in mice) and decreases DNMT1 activity [37]. We found that overexpression of wild-type DNMT1 and a DNMT1S717A mutant both increased genome-wide 5 mC levels, but the effect of the latter was more significant. Thus, our results showed that ERK regulates the subcellular localization of DNMT1 by phosphorylating DNMT1 S717, thereby regulating genome-wide 5 mC levels.

Previous studies from our research group found that PGC7 is a native unfolded protein that exerts different functions by binding to different functional proteins [38]. In addition, it is also known that PGC7 regulates the expression of many genes [16,39]. Therefore, we speculated that PGC7 may affect ERK activity by interacting with the ERK protein or by directly regulating the expression of ERK or its upstream kinases. Our RT-qPCR results identified a significant decrease in the transcription of MEK1 (i.e., an upstream kinase of ERK1/2) in *Pgc7*-knockdown F9 cells. This suggests that the downregulation of *Mek1* transcription by PGC7 may be a pathway leading to the downregulation of ERK phosphorylation.

Active demethylation mediated by Tet has been found to promote the expression of PGC7 in mESCs, while PGC7 mediates DNA demethylation by directly binding to UHRF1 to promote its release from chromatin [40]. Li et al. found that knocking out *Pgc7* resulted in the ectopic nuclear accumulation of UHRF1 and the localization of DNMT1 in the nucleus, leading to abnormal DNA methylation of major satellite repeats during oogenesis [14]. These studies suggest that PGC7 affects overall DNA methylation by regulating the binding of UHRF1 to chromatin or by altering the subcellular localization of UHRF1. However, we found that endogenous DNMT1 tended to be located in the nucleus of *Pgc7*-knockdown F9 cells and that the level of ERK phosphorylation decreased significantly, but the localization of UHRF1 did not significantly change. Combined with the nuclear translocation of DNMT1 induced by PD treatment, we suggest that the regulation of DNMT1 nuclear translocation by PGC7 may be mediated by ERK rather than by UHRF1 localization.

DNA hypermethylation and global DNA hypomethylation in promoter regions are frequently detected in human cancers, and genomic DNA hypomethylation is common in a variety of gastrointestinal diseases. Moreover, DNMT1 also plays a major role in gastrointestinal development and diseases [28,41,42]. Studies of disease and drug therapy have found that ERK signaling plays a critical role by regulating DNMT1 expression [23]. Therefore, our identification of the mechanism by which PGC7 regulates DNA methylation, i.e., via phosphorylation of DNMT1 at ser717 by ERK, may have implications for the treatment of DNA methylation-related diseases. The proposed mechanism of action is shown in Figure 5e. Knockdown of *Pgc7* results in downregulation of ERK phosphorylation, which affects phosphorylation of DNMT1 at ser717 by ERK, and therefore causes DNMT1 to translocate to the nucleus, thereby increasing genome-wide DNA methylation levels. Thus, in this study, we propose a novel mechanism by which PGC7 regulates genome-wide DNA methylation via phosphorylation of DNMT1 at ser717 by ERK. These findings also necessitate the re-evaluation of the detailed mechanism by which PGC7 regulates DNA methylation in the development of ESCs.

## 4. Materials and Methods

### 4.1. Cell Culture

HEK-293T cells, NIH3T3 cells, and F9 embryonal carcinoma (EC) cells were obtained from the American Type Culture Collection (ATCC, Manassas, VA, USA), cultured in Dulbecco’s modified Eagle’s medium containing 10% fetal bovine serum (Gibco, Waltham, MA, USA), and incubated at 37 °C in a humidified incubator (Thermo Fisher, Waltham, MA, USA) with 5% CO_2_. Cells were transfected with siRNAs or plasmids or treated with PD0325901 (Selleck Chemicals, Boston, MA, USA) at a final concentration of 1μM for 12h. For all PD treatments, DMSO was used as a negative control.

### 4.2. Kinase Phosphorylation Target Prediction

Kinase phosphorylation sites were queried in the Scansite (https://scansite4.mit.edu/, accessed on 15 September 2022) and PhosphoSitePlus (https://www.phosphosite.org/homeAction.action, accessed on 15 September 2022) phosphorylation mass spectrometry databases. Sites were predicted using GPS5.0 software to screen candidate DNMT1 phosphorylation sites.

### 4.3. Plasmids and Site-Directed Mutants

cDNA encoding mouse *Pgc7* was cloned into p3×Flag-CMV-10, pCMV-HA, and pEGFP-C1 vectors. cDNAs encoding mouse Dnmt1, Mek1, and Erk1/2 were separately cloned into p3×Flag-CMV-10 vectors. cDNA encoding mouse Erk1 was also cloned into pCMV-HA. Site-directed mutants of mouse Dnmt1 were amplified via overlapping extension PCR and were then cloned into p3×Flag-CMV-10 vectors. Truncated forms of Dnmt1 and its S717A mutant were also cloned into p3×Flag-CMV-10 vectors. The sequences of recombinant expression vectors and mutants were confirmed by sequencing. All primers used are listed in Appendix A.

### 4.4. Transfection

Specific siRNA oligonucleotides for *Pgc7* with green fluorescence (named FAM) were synthesized by GenePharma, then transfected into F9 cells for 36 h using Lipofectamine 2000 (Invitrogen, Waltham, MA, USA), applied according to the manufacturer’s protocol. Si*Pgc7* -1, a sequence with a stronger knockdown effect, was selected for all experiments. All sequences used are listed in Appendix A. Recombinant expression vectors and mutants were transfected into F9, HEK293T, or NIH3T3 cells as indicated in the above method.

### 4.5. Methylation and Immunofluorescence Staining

For methylation staining, cells were cultured normally. For immunofluorescence staining, round coverslips were first laid out and coated with gelatin before cells were added. F9, HEK293T, or NIH3T3 cells were washed with 1×PBS and fixed in Immunol staining fix solution (Beyotime, Shanghai, China) for 15 min at room temperature. Cells were then washed for 5 min three times with 1×PBS, then permeabilized with 0.1% Triton X-100 for 15 min. Next, cells were again washed in PBS as before, then treated with RNase A at a final concentration of 50 μg/mL for 1h in a dark room. The cells were washed one final time, then treated with 4 N HCl for 5 min. After this application, the cells were neutralized with Tris-HCl (pH 8.5) for 10 min. The cells were again triple washed in PBS before being blocked in QuickBlock™ blocking buffer for Immunol staining (Beyotime, Shanghai, China) for 15 min at room temperature or at 4 °C overnight. After another triple PBS wash, cells were incubated with mouse anti-5-methylcytosine (5 mC) antibody (1:500, Active Motif, Shanghai, China), rabbit anti-DNMT1 antibody (1:500, Abcam, Cambridge, UK), anti-ERK antibody (1:800, Cell Signaling Technology, Boston, MA, USA), antiphospho-ERK antibody (1:400, Cell Signaling Technology, Boston, MA, USA), mouse anti-Flag antibody (1:800, Sigma Aldrich, Saint Louis, MO, USA), or anti-UHRF1 antibody (1:200, Santa Cruz Biotechnology, USA) at 4 °C overnight. Cells were then triple washed before being incubated with appropriate secondary antibodies (1:500, Thermo Scientific, Waltham, MA, USA) for two hours at room temperature. The cells were then triple washed once again before being stained with DAPI for ten minutes. After another wash, images were captured using a fluorescence microscope. For immunofluorescence staining, the step after permeabilization was blocking, and the remaining steps were the same as those for methylation staining.

### 4.6. Immunoprecipitation and Western Blotting

To obtain whole cell extracts for immunoprecipitation, cells were first lysed with IP lysis buffer (Pierce, Thermo Fisher, Waltham, MA, USA; added with a proteinase inhibitor cocktail or both a proteinase inhibitor cocktail and a protein phosphatase inhibitor) for 20 min at 4 °C. After centrifugation at 12,000× *g* for 15 min, 10% of the supernatant was taken as input, and the rest was divided equally into IgG and IP treatments or (as appropriate) only for an IP treatment. Next, the appropriate IgG or IP was incubated with respective antibodies at 4 °C overnight. After incubation, protein A/G plus agarose beads were added to capture corresponding proteins. After a triple wash in IP wash buffer for 5 min each time, immunoprecipitated proteins were then boiled with 1×SDS for 10 min. The samples were then analyzed by Western blot. To obtain whole cell extracts for normal Western blots, cells were first lysed with RIPA buffer (high) (Solarbio, Beijing, China). This buffer was also added either with proteinase inhibitor cocktail or with a combination of proteinase inhibitor cocktail and a protein phosphatase inhibitor. Lysis took place at 4 °C for 20 min.

After centrifugation at 12,000× *g* for 15 min, the supernatant was boiled with 5×SDS diluted to a final concentration of 1× for 10 min. Next, the samples were analyzed by Western blot. To do so, the protein samples were first separated by SDS/PAGE then transferred to a PVDF membrane. The membranes were then blocked with 10% skimmed milk or 5% BSA for 1–2 h before being incubated with anti-ERK antibody (1:1000, Cell Signaling Technology, Boston, MA, USA), antiphospho-ERK antibody (1:1000, Cell Signaling Technology, Boston, MA, USA), anti-PGC7 antibody (1:1000, Abcam ab19878, Cambridge, UK), anti-GAPDH antibody (1:1000, TransGen Biotech, Beijing, China), anti-DNMT1 antibody (1:1000, Cell Signaling Technology D63A6, Boston, MA, USA), anti-Flag antibody (1:1000, Sigma, Saint Louis, MO, USA), anti-HA antibody (Sigma, Saint Louis, MO, USA for IP; 1:1000, TransGen Biotech, Beijing, China for Western blot), and anti-GFP antibody (1:1000, Active Motif, Shanghai, China) at 4 °C overnight. After three 10 min washes using 1×TBST, the membranes were incubated with HRP-labeled goat antirabbit or antimouse IgG(H+L) (1:1000 or 1:2000, Beyotime, Shanghai, China), IPKine HRP AffiniPure goat antimouse IgG light chain (1:3000, AmyJet Scientific, Wuhan, China), or goat anti-rabbit IgG heavy chain (1:10000, AmyJet Scientific, Wuhan, China) for 1–2 h at room temperature. Finally, detection was performed using a ChemiDoc Touch imaging system (Bio-rad, Hercules, CA, USA).

### 4.7. Genome Extraction and Dot Blot Analysis

Cells transfected with siRNAs or plasmids were first washed in 1×PBS. Cells were then harvested and resuspended in a liquid mixture containing TE buffer, cell lysis solution, and proteinase K. After shaking incubation at 56 °C for 10 min, the lysate was incubated with RNase A for 10 min at room temperature. A DNA extraction solution was then added, and the mixture was shaken before being incubated for 15 min at room temperature. After centrifugation at maximum speed for 10 min, the supernatant was collected, and DNA was precipitated at −20 °C by adding isopropanol and leaving overnight. After centrifugation at 12,000× *g* for 15 min, the DNA pellet was washed with 75% ethanol, air dried, and then resuspended in the elution buffer.

DNA fragments of different sizes but the same volume were incubated in a PCR incubator at 95 °C for 10 min, after which an aliquot of ammonium acetate of the same volume was added, and the mixture was then placed on ice. The NC membrane was then immersed in 6×SSC (saline sodium citrate) buffer before DNA transfer. The NC membrane was then washed in 2×SSC buffer and incubated at 50 °C in an oven for 30 min. After UV crosslinking for 30 min, the membrane was blocked with 5% skimmed milk for 1–2 h. The NC membrane was then incubated with mouse anti-5 mC antibody (1:4000) at 4 °C overnight after being washed three times with 1×TBST for ten minutes each time. Finally, the NC membrane was incubated with HRP-labeled goat antimouse IgG(H+L) (1:1000, Beyotime) for 1–2 h at room temperature and fluorescence detection was performed using a ChemiDoc Touch imaging system (Bio-rad, Hercules, CA, USA).

### 4.8. RNA Extraction and RT-qPCR

F9 cells were harvested and total RNA was extracted using a TransZol Up kit. Total RNA was then converted to cDNA using a reverse transcription kit (TransGen Biotech, Beijing, China), with all procedures performed according to the manufacturer’s protocol. RT-qPCR analysis was performed using TB Green Premix Ex Taq II (Tli RNaseH Plus) (Takara, Beijing, China) and a CFX96 Real-Time System (Bio-Rad, Hercules, CA, USA). Samples were normalized to GAPDH. All primers used for RT-qPCR were designed using the NCBI online primer design tool and are listed in Appendix A.

### 4.9. Statistical Analysis

All data were presented as means ± SD for quantitative analysis. Mean differences between the treated and control groups associated with *p*-values of <0.05 were considered to be statistically significant. All statistical testing used Student’s *t*-tests as implemented in GraphPad Prism 6 software. ImageJ 1.49 software was used to analyze the gray value of the Western blotting and fluorescence intensity. Original collected data are shown in the Appendix A.

## 5. Conclusions

In this study, we first identified a mode by which PGC7 regulates genome-wide DNA methylation by maintaining ERK phosphorylation. The ser717 residue (S717) of the DNMT1 protein is regulated by ERK phosphorylation, and phosphorylation modification at this site affects the intracellular localization of DNMT1. Knockdown of PGC7 in F9 cells led to the inhibition of ERK phosphorylation, which resulted in the accumulation of DNMT1 in the nucleus. Moreover, knockdown of PGC7 or inhibition of ERK phosphorylation was found to increase genome-wide DNA methylation levels. In conclusion, we discovered a novel pathway by which PGC7 regulates genome-wide DNA methylation via regulating the activity of a key signaling pathway along the PGC7-ERK-DNMT1S717 axis.

## Figures and Tables

**Figure 1 ijms-24-03093-f001:**
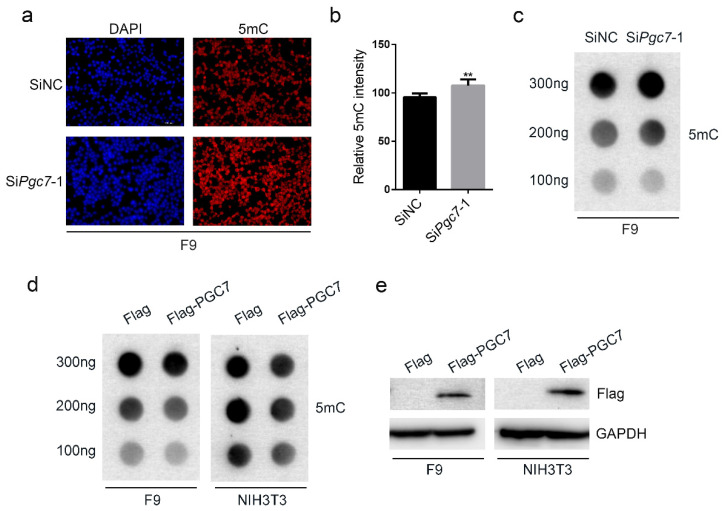
PGC7 regulates genome-wide DNA methylation. (**a**,**b**) Knockdown of *Pgc7* in F9 cells increased genome-wide 5 mC levels. F9 cells were transfected with SiNC or Si*Pgc7*-1 for 36h, after which a methylation staining experiment was performed. ImageJ 1.49 software was used to determine 5 mC fluorescence intensity from five or more images for both the SiNC and Si*Pgc7*-1 groups. ** *p* < 0.01. (**c**) Knockdown of *Pgc7* in F9 cells increased genome-wide 5 mC levels. SiNC or Si*Pgc7*-1 was transfected into F9 cells for 36 h. Genomic material was then extracted, and dot blot experiments were performed. (**d**,**e**) Overexpression of PGC7 in F9 and NIH3T3 cells reduced genome-wide 5 mC levels. F9 and NIH3T3 cells were transfected with p3×Flag-CMV-10 or p3×Flag-CMV-10-PGC7 for 36–48 h, respectively. (**d**) Genomic material was then extracted, and dot blot experiments were then performed. (**e**) Western blot analysis of the efficiency of PGC7 overexpression.

**Figure 2 ijms-24-03093-f002:**
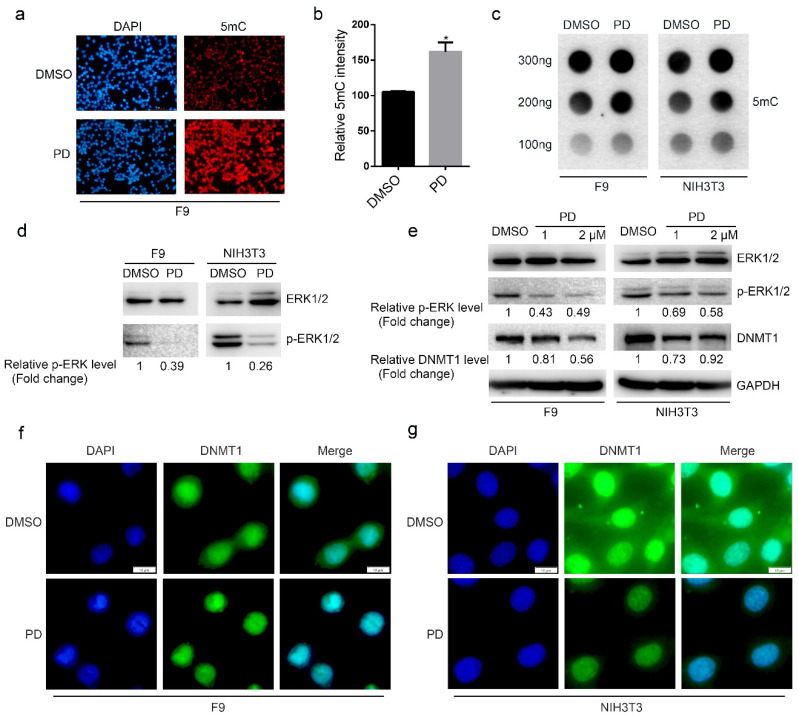
ERK regulates genome-wide DNA methylation. (**a**,**b**) PD treatment increased genome-wide 5 mC levels in F9 cells. F9 cells were treated with 1μM PD for 12 h, after which a methylation staining experiment was performed. ImageJ 1.49 software was used to determine 5 mC fluorescence intensity, and three images were analyzed for both the DMSO and PD groups. * *p* < 0.05. (**c**) PD treatment increased genome-wide 5 mC levels in F9 and NIH3T3 cells. F9 and NIH3T3 cells were treated with 1 μM PD for 12 h, after which genomic DNA was extracted and a dot blot experiment was performed. (**d**) Western blot results showing the effect of PD treatment. (**e**) PD treatment downregulated the expression of DNMT1. F9 and NIH3T3 cells were treated with 1 or 2 μM PD for 12 h, respectively, after which total protein samples were obtained for Western blotting. (**f**,**g**) PD treatment promoted the nuclear localization of endogenous DNMT1 in F9 and NIH3T3 cells. F9 and NIH3T3 cells were treated with 1 μM PD for 12 h, and immunofluorescence staining was performed.

**Figure 3 ijms-24-03093-f003:**
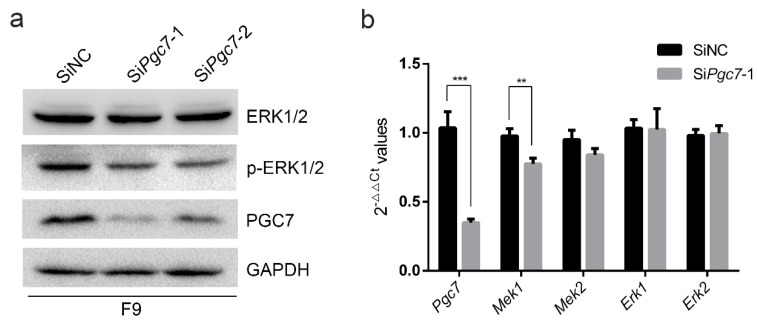
PGC7 regulates ERK activity by regulating *Mek1* transcription. (**a**) Knockdown of PGC7 resulted in the downregulation of ERK phosphorylation in F9 cells. F9 cells were transfected with SiNC or Si*Pgc7* for 36h, after which a Western blot was performed. (**b**) The transcription levels of *Mek1*, *Mek2*, *Erk1*, and *Erk2* in F9 cells in which *Pgc7* was knocked down. F9 cells were transfected with SiNC or Si*Pgc7*-1 for 36 h prior to the extraction of total RNA. RT-qPCR was performed after reverse transcription. Data are presented as means ± SD for quantitative analysis. *** *p* < 0.001, ** *p* < 0.01.

**Figure 4 ijms-24-03093-f004:**
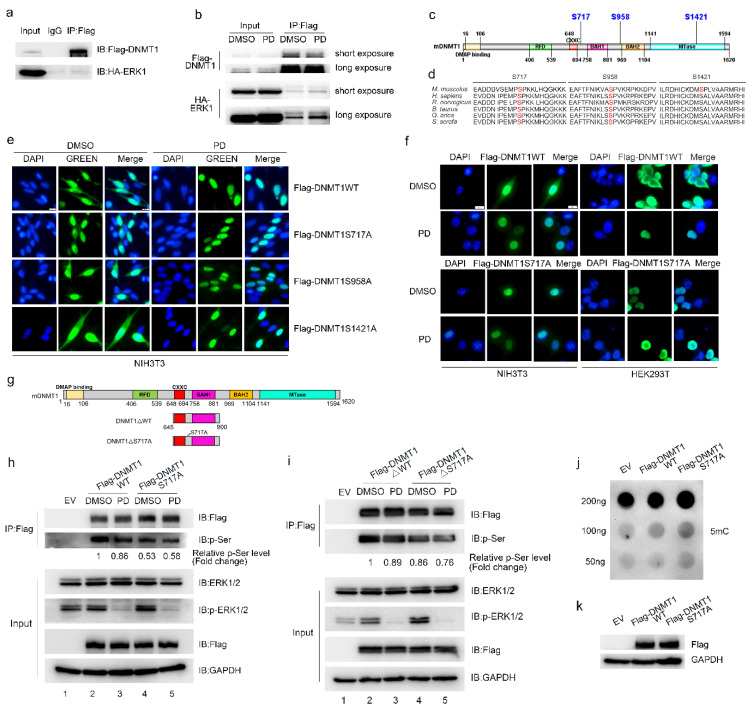
Phosphorylation of DNMT1 at S717 by ERK regulates the subcellular localization of DNMT1. (**a**,**b**) ERK1 interacts weakly with DNMT1 in HEK293T cells. PD treatment enhances the interaction between ERK and DNMT1. Here, HEK293T cells were co-transfected with p3×Flag-CMV-10-DNMT1 and pCMV-HA-ERK1, and the Flag-DNMT1 fusion protein was captured by an anti-Flag antibody. Mouse IgG was used as a negative control. A Western blot was used to visualize the expression of DNMT1 and ERK1 by measuring fluorescent signals linked to anti-Flag and anti-HA antibodies, respectively. (**c**) Schematic diagram of mouse DNMT1 protein domains and potential ERK phosphorylation targets as predicted by GPS5.0 software. (**d**) Evolutionary conservation comparison of sequences near the S717, S958, and S1421 residues of the DNMT1 protein from different species. (**e**) Subcellular localization of wild-type DNMT1 (DNMT1WT) and the DNMT1S717A, DNMT1S958A, and DNMT1S1421A mutant proteins in NIH3T3 cells following DMSO and PD treatment. NIH3T3 cells were transfected with p3×Flag-CMV-10-DNMT1WT, p3×Flag-CMV-10-DNMT1S717A, p3×Flag-CMV-10-DNMT1S958A, and p3×Flag-CMV-10-DNMT1S1421A for 24h. Transfected cells were then treated with 1 μM PD for 12 h, after which immunofluorescence staining was performed. (**f**) Subcellular localization of wild-type DNMT1 and DNMT1S717A mutant protein following DMSO and PD treatment in NIH3T3 and HEK293T cells. NIH3T3 and HEK293T cells were transfected with p3×Flag-CMV-10-DNMT1WT and p3×Flag-CMV-10-DNMT1S717A for 24 h, respectively. Transfected cells were then treated with 1 μM PD for 12 h, after which immunofluorescence staining was performed. (**g**) Schematic diagram of the domains of truncated DNMT1WT and DNMT1S717A. (**h**,**i**) The levels of overall serine phosphorylation after DMSO and PD treatment in HEK293T cells that overexpressed wild-type DNMT1 and DNMT1S717A mutant protein. HEK293T cells were transfected with p3×Flag-CMV-10, p3×Flag-CMV-10-DNMT1WT, and p3×Flag-CMV-10-DNMT1S717A or p3×Flag-CMV-10, p3×Flag-CMV-10-DNMT1△WT, and p3×Flag-CMV-10-DNMT1△S717A, respectively. Flag-DNMT1WT, Flag-DNMT1S717A, Flag-DNMT1△WT, and Flag-DNMT1△S717A fusion proteins were captured by the anti-Flag antibody for visualization. Mouse IgG was used as a negative control. A Western blot was used to visualize DNMT1 expression and the levels of overall serine phosphorylation by anti-Flag and anti-p-Ser antibodies, respectively. (**j**) Genome-wide 5 mC levels in HEK293T cells overexpressing wild-type DNMT1 and DNMT1S717A mutant protein. HEK293T cells were transfected with p3×Flag-CMV-10, p3×Flag-CMV-10-DNMT1WT, and p3×Flag-CMV-10-DNMT1S717A, respectively. Total genomic DNA was then extracted and a dot blot assay was performed. (**k**) The overexpression efficiency of wild-type DNMT1 and the DNMT1S717A mutant as determined by Western blot.

**Figure 5 ijms-24-03093-f005:**
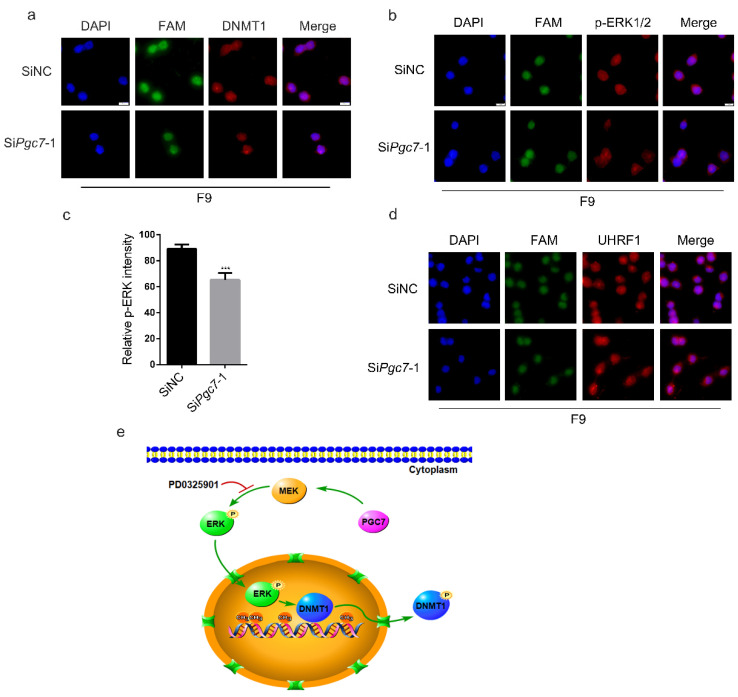
PGC7 regulates the subcellular localization of DNMT1 via ERK. SiNC and Si*Pgc7*-1 were transfected into F9 cells for 36 h, after which immunofluorescence staining was performed. (**a**) Knockdown of *Pgc7* promoted the nuclear localization of endogenous DNMT1. (**b**,**c**) Knockdown of *Pgc7* resulted in the downregulation of ERK phosphorylation. 5 mC fluorescence intensity analysis was performed using ImageJ 1.49 software. Three or more replicate images were analyzed for both the SiNC and Si*Pgc7*-1 groups. *** *p* < 0.001. (**d**) Knockdown of *Pgc7* did not affect the subcellular localization of endogenous UHRF1. (**e**) Schematic diagram of the proposed regulatory mechanism.

## Data Availability

The data supporting the findings of this study are available within the article and Appendix A.

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
