# Peer review of "PGC7 Regulates Genome-Wide DNA Methylation by Regulating ERK-Mediated Subcellular Localization of DNMT1"

_ijms, 2023, doi:10.3390/ijms24043093_

Round 1

Reviewer 1 Report

The paper explores some mechanisms by which global DNA methylation is regulated by ERK. The authors found that PGC7 regulates global DNA methylation by phosphorylation of DNMT1 at Ser717 by ERK. It is an important study for the field and merits publication, with some concerns.

The authors need to briefly explain in the manuscript why they choose the F9 cell line to conduct the study.

My main concern is regarding the western blot and immunofluorescence. Why authors didn’t quantify all western blots and IF performed? This needs to be done. Also, how many samples and replicates did the authors performed to be statistically significant?

The figure 1e is confusing. What is EV? Why there is Flag-PGC7 above and next to the band? You need to show what is the antibody used next to the band. The bands are showing increase in methylation of Flag-PGC7 here? Which is the opposite that we describe in the result section (page 3, line 107).

Figure 2d and e is showing increase in ERK after treatment with PD in NIH3T3, which is contradictory. Again, please quantify the western blots and describe the sample size.

Figure 4 is a mess. With d jumping to g and back to e etc. 4b is showing co-localization of DNMT1 and ERK in the nucleus, and not cytoplasm as described in results (Page 6, line 187). Again, important to quantify all wb and IF.

Author Response

Reviewer1:

The paper explores some mechanisms by which global DNA methylation is regulated by ERK. The authors found that PGC7 regulates global DNA methylation by phosphorylation of DNMT1 at Ser717 by ERK. It is an important study for the field and merits publication, with some concerns.

Comment 1: The authors need to briefly explain in the manuscript why they choose the F9 cell line to conduct the study.

Response 1: The reason why we choose the F9 cell line to conduct the study is that the F9 cell line, in which the PGC7 expression and ERK activity is high, is derived from a mouse testicular teratoma that originated from pluripotent germ cells. It is used as a model for cell differentiation research because of its ability to differentiate into endodermal-like derivatives after treatment with retinoic acid. Moreover, because of its limited differentiation ability, F9 EC cells can also be considered as representative of low pluripotent stem cells. Given that F9 EC cell line is derived from testicular teratomas, it is a useful tool for investigating pluripotency, differentiation, and tumorigenesis.

Comment 2: My main concern is regarding the western blot and immunofluorescence. Why authors didn’t quantify all western blots and IF performed? This needs to be done. Also, how many samples and replicates did the authors performed to be statistically significant?

Response 2: Thanks for your suggestion. We have quantified western blots and IF performed in the manuscript. Three samples and three replicates were performed to be statistically significant. This result is shown in the updated manuscript.

Comment 3: The figure 1e is confusing. What is EV? Why there is Flag-PGC7 above and next to the band? You need to show what is the antibody used next to the band. The bands are showing increase in methylation of Flag-PGC7 here? Which is the opposite that we describe in the result section (page 3, line 107).

Response 3: Thanks for your suggestion. We are sorry to not indicate the meaning of EV. EV means empty vector, that is p3×Flag-CMV-10 in our study, and we have revised it in figure 1e and other figures involved.

Flag-PGC7 above means the overexpression of Pgc7 in cells, and Flag-PGC7 next to the band means what the band is. We have showed what was the antibody used next to the band in manuscript. As mentioned above, the bands are showing the overexpression of Flag-PGC7 fusion protein to check transfection in figure 1d, and the result of figure 1d showed that overexpression of PGC7 reduced the genome-wide 5mC levels.

Comment 4: Figure 2d and e is showing increase in ERK after treatment with PD in NIH3T3, which is contradictory. Again, please quantify the western blots and describe the sample size.

Response 4: Thanks for your suggestions. We have quantified the western blots in figure 2d and e. This result is shown in the updated manuscript

Comment 5: Figure 4 is a mess. With d jumping to g and back to e etc. 4b is showing co-localization of DNMT1 and ERK in the nucleus, and not cytoplasm as described in results (Page 6, line 187). Again, important to quantify all wb and IF.

Response 5: Thanks for your suggestion. We are sorry for no good typesetting. We have rearranged figure 4 in the updated manuscript.We have quantified western blots and IF. Thank you a million for your comments and suggestions, and this result is shown in the updated manuscript.

Reviewer 2 Report

In this manuscript, Wei et al, have analyzed the involvement of Pgc7 in DNA methylation regulation focusing on the post-translational modification of DNMT1. The authors demonstrated that i) Pgc7 knockdown or inhibition of ERK activity increased genome-wide DNA methylation, ii) inhibition of ERK activity accumulated DNMT1 in the nucleus, iii) ERK phosphorylated DNTM1 at Ser717 which was important for DNMT1 localization, and iv) Pgc7 knockdown downregulated ERK phosphorylation and promoted DNMT1 accumulation in the nucleus. Based on these observations, the authors concluded that Pgc7 regulates genome-wide DNA methylation through the phosphorylation of DNMT1 at Ser717 by ERK. 

However, the provided data is not convincing enough to support the authors' conclusions.

Major Comments: 

1. The authors should establish and analyze stable knockdown cells of Pgc7 using multiple shRNAs. The data of Pgc7 could not convey the authors’ conclusions because of the following reasons.

a) The data of immunofluorescence and dot-blot was not clear (Figures 1a, b, and c, and Figures 5a, b, and d).

b) It is impossible to know which cells were actual knockdown cells without Pgc7 staining (Figure 1a and b, and Figure 5a, b, and d). 

c) Most transfection reagents have a cytotoxic effect, it is not appropriate to analyze the change of cellular phenotype after siRNA transient transfection. 

2. Figures 4a and b. The data of co-IP and immunofluorescence staining could not convey the authors’ conclusions because of the following reasons.

a) The intensity of the band of HA-Erk1 was very weak after IP using flag-tagged antibody meaning no interaction between flag-Dnmt1 and HA-Erk1. 

b) The resolution of the images is low to conclude the subcellular co-localization of Erk1/2 and Dnmt1.

3. Figure 1d and e. F9 cells were used both for knockdown and overexpression of Pgc7. The authors should use a cell line with low endogenous expression of Pgc7 for overexpression. Also, the authors should use the vector expressing a control gene such as GFP instead of the empty vector.

4. Although the authors often explained the data of dot-blot using the expressions “significantly increased” or “significantly reduced”, the data of dot-blot can not be used for statistical analysis (Figure 1c and d, Fig. 2c). 

5. Fig. 3b. The expression levels should be represented as actual copy numbers or actual delta-delta Ct values, not "relative mRNA levels".

Minor Comments:

1. The authors should carefully edit the entire manuscript and also consider utilizing professional English proofreading. Especially, the Discussion section should be edited to reduce wordiness and redundancy and make it more comprehensive.

2. The status of mycoplasma infection should be analyzed and described in the Materials and Methods section.

Author Response

Reviewer 2

In this manuscript, Wei et al, have analyzed the involvement of Pgc7 in DNA methylation regulation focusing on the post-translational modification of DNMT1. The authors demonstrated that i) Pgc7 knockdown or inhibition of ERK activity increased genome-wide DNA methylation, ii) inhibition of ERK activity accumulated DNMT1 in the nucleus, iii) ERK phosphorylated DNTM1 at Ser717 which was important for DNMT1 localization, and iv) Pgc7 knockdown downregulated ERK phosphorylation and promoted DNMT1 accumulation in the nucleus. Based on these observations, the authors concluded that Pgc7 regulates genome-wide DNA methylation through the phosphorylation of DNMT1 at Ser717 by ERK. 

However, the provided data is not convincing enough to support the authors' conclusions.

Major Comments: 

Comment 1. The authors should establish and analyze stable knockdown cells of Pgc7 using multiple shRNAs. The data of Pgc7 could not convey the authors’ conclusions because of the following reasons.

  1. a) The data of immunofluorescence and dot-blot was not clear (Figures 1a, b, and c, and Figures 5a, b, and d).
  2. b) It is impossible to know which cells were actual knockdown cells without Pgc7 staining (Figure 1a and b, and Figure 5a, b, and d). 
  3. c) Most transfection reagents have a cytotoxic effect, it is not appropriate to analyze the change of cellular phenotype after siRNA transient transfection. 

Response  

Thank you very much for your comments. We have tried many times to establish Pgc7 knockout cell lines, but failed. Pgc7 knockout cell lines cannot survive. Later, we constructed a stable Pgc7 knockdown cell line through lentivirus transfection. However, the knockdown cell line was in poor condition and lacked cell vitality, especially the irregular shape of cells and nuclei, which seriously interfered with the subsequent detection of the localization of DNMT1 and other proteins. Therefore, we knocked down Pgc7 by transfecting two small interfering RNAs targeting different positions of Pgc7. Through observation with FAM-NC, we found that the transfection efficiency of small interfering RNAs reached more than 90%, Also, we confirmed that the interference efficiency of two small interfering RNAs targeting different positions of Pgc7 reached more than 65% through RT-qPCR, compare with stable knockdown cells of Pgc7, the status of F9 cells is also in good condition, especially the morphology of cells and nuclei, which is conducive to the subsequent protein location detection. Therefore, we use the method of transient small interfering RNA to knock down Pgc7 in the subsequent experiment.

In addition, the phenotype we observed is the change of DNA methylation, which is a rapid change at the epigenetic regulation level. Therefore, the interference of transient Pgc7 is also conducive to detecting the changes of protein location and DNA methylation in cells at a short time.

Response a) We have readjusted the image contrast and definition of Figure1a、1b、and c, and Figure 5a、 b and d. At the same time, we added the transfection efficiency detection pictures of FAM-NC and FAM-siPgc7 in the manuscript to prove the transfection efficiency of small interfering RNA. This figure is marked as FAM.

Response b) Through observation with FAM-NC, we found that the transfection efficiency of siRNAs reached more than 90%. At the same time, we confirmed that the interference efficiency of two siRNAs targeting different positions of Pgc7 reached more than 65% by RT-qPCR. Therefore, we think that cells in the field of vision can represent cells with knockdown of Pgc7.

Response c) After transient transfection using siRNA for 6h, we changed the fresh culture medium to continue to culture cells for 24h, and then collected the cells. Compared with the cells not transfected, the change of cellular phenotype after transient transfection using siRNA has not changed in detail.

Comment 2. Figures 4a and b. The data of co-IP and immunofluorescence staining could not convey the authors’ conclusions because of the following reasons.

  1. a) The intensity of the band of HA-Erk1 was very weak after IP using flag-tagged antibody meaning no interaction between flag-Dnmt1 and HA-Erk1. 
  2. b) The resolution of the images is low to conclude the subcellular co-localization of Erk1/2 and Dnmt1.

Response a) Thank you very much for your suggestion. According to your opinion, we repeated the experiment in Figures 4a. At the same time, we added PD treatment to detect whether the interaction between non-phosphorylated ERK and DNMT1 is enhanced. The experimental results showed that there was also a weak interaction between ERK-DNMT1. At the same time, the interaction between non-phosphorylated ERK and DNMT1 was enhanced after PD treatment to inhibit ERK phosphorylation. We have marked the new experimental results as Figures 4b, and the corresponding illustrations have also been changed.

Response b) Thank you very much for your suggestion. We have readjusted the contrast and definition of the pictures in Figures 4b. the subcellular co-localization of ERK1/2 and DNMT1 was clearer.

Comment 3. Figure 1d and e. F9 cells were used both for knockdown and overexpression of Pgc7. The authors should use a cell line with low endogenous expression of Pgc7 for overexpression. Also, the authors should use the vector expressing a control gene such as GFP instead of the empty vector.

Response 3. Thank you for your suggestion. we detected overexpression in NIH3T3 cells, with low endogenous expression of PGC7, and the experimental trend was consistent with that of F9 cells with high expression of PGC7. After overexpression of PGC7 in NIH3T3 cells, we observed more significant changes in the modification of DNA methylation. Secondly, we also constructed the vector of GFP-PGC7 in the preliminary experiment, and the results were consistent. Since the molecular weight of Flag tag is very small, which is conducive to improving the transfection efficiency. In this study, we adopted p3×Flag-CMV-10 is used as a control gene.

Comment 4. Although the authors often explained the data of dot-blot using the expressions “significantly increased” or “significantly reduced”, the data of dot-blot cannot be used for statistical analysis (Figure 1c and d, Fig. 2c). 

Response 4. Thank you for your suggestion. We have removed this description, and replaced it with another description. This result is shown in the updated manuscript.

Comment 5. Fig. 3b. The expression levels should be represented as actual copy numbers or actual delta-delta Ct values, not "relative mRNA levels".

Response  5. Thank you for your suggestion. We use actual delta-delta Ct values to process data, but the mark is wrong, which have been corrected in Fig.3b.

Minor Comments:

Comment 1. The authors should carefully edit the entire manuscript and also consider utilizing professional English proofreading. Especially, the Discussion section should be edited to reduce wordiness and redundancy and make it more comprehensive.

Response 1. Thank you for your suggestion. The Discussion section had been edited to reduce wordiness and redundancy. This result is shown in the updated manuscript.

Comment 2. The status of mycoplasma infection should be analyzed and described in the Materials and Methods section.

Response 2. Thank you for your suggestion. Our initial cells were tested for mycoplasma, and there was no mycoplasma contamination.

Round 2

Reviewer 1 Report

I cannot see the updates that the authors mentioned in the updated manuscript, i.e. the western blot and immunofluorescence quantifications.

Author Response

Response:

Thanks a million for your comments. We have updated the quantitative analysis of western blotting and immunofluorescence to the materials and methods (page 14, line 541-543), and the detailed original collected data were shown in the Supplementary Table 1-8 and marked them with red font.

Reviewer 2 Report

Wei et al. have added some new experiments and descriptions in the revised manuscript to address the Reviewer’s comments. However, there are specific issues that the authors should modify for acceptance.  

Major comments:

1. Response b) to Comment 1. Although the authors mentioned that the transfection efficiency of siRNAs reached more than 90%, it is impossible to distinguish whether the FAM-siRNAs were really into the cells or bound to the cellular membrane under a normal microscope and the images taken at low magnification. Also, RT-qPCR showed the interference efficiency of two Pgc7 siRNA was at least 65%, meaning that 35% of cells have intact Pgc7 or only 65% of mRNA was reduced. To conclude that Pgc7 knockdown downregulated ERK phosphorylation and promoted DNMT1 accumulation in the nucleus, the authors should perform immunostaining of Pgc7 and show the changes of ERK and DNMT1 in the cells with no Pgc7 expression. 

Also, the authors mentioned “two small interfering RNAs targeting different positions of Pgc7” in the author’s reply, all data seems to be obtained using only one siRNA, and the sequence of only one siPgc7 was shown in Supplementary Table 2.

2. Response b) to Comment 2. Although the authors have readjusted the contrast and definition of the pictures in Figure 4b, it is impossible to conclude specific co-localization of ERK1/2 and DNMT1 in cytoplasm and nucleus because all proteins localize in cytoplasm or nucleus. The authors should remove these figures, change the expressions, or perform additional experiments such as Proximity Ligation Assay to show the interaction between two molecules.

Minor comments:

The authors did not explain “FAM” in the main text.

Author Response

Major comments:

  1. Response b) to Comment 1. Although the authors mentioned that the transfection efficiency of siRNAs reached more than 90%, it is impossible to distinguish whether the FAM-siRNAs were really into the cells or bound to the cellular membrane under a normal microscope and the images taken at low magnification. Also, RT-qPCR showed the interference efficiency of two Pgc7 siRNA was at least 65%, meaning that 35% of cells have intact Pgc7 or only 65% of mRNA was reduced. To conclude that Pgc7 knockdown downregulated ERK phosphorylation and promoted DNMT1 accumulation in the nucleus, the authors should perform immunostaining of Pgc7 and show the changes of ERK and DNMT1 in the cells with no Pgc7 expression. 

Also, the authors mentioned “two small interfering RNAs targeting different positions of Pgc7” in the author’s reply, all data seems to be obtained using only one siRNA, and the sequence of only one siPgc7 was shown in Supplementary Table 2.

Response 1: Thanks a million for your comments. This is a very rigorous and important question. But we think our experimental results are credible. Firstly, it can be seen by FAM, the interference efficiency of two Pgc7 siRNA was at least 65% do not means that 35% of cells have intact Pgc7 mRNA, but was about 90% cells have reduced at least 75% Pgc7 mRNA. Secondly, we randomly took at least four different fields of view between siNC and SiPgc7, and the results showed that all DNMT1 localization tended to be in the nucleus in the SiPgc7 group. Second, regarding your experiments with endogenous PGC7 staining, we did the same thing. However, since the PGC7 (anti-PGC7 antibody Abcam ab19878)) and DNMT1(anti-DNMT1 antibody Cell Signaling Technology D63A6)) antibodies are rabbit derived,it is not possible to detect the knockdown efficiency of PGC7 in the same field of view by the IF method, so we indirectly detected the knockdown efficiency of PGC7 by FAM. The knockdown efficiency of endogenous PGC7 alone was measured, and the results shown below show that after transfection with SiPgc7, the amount of endogenous PGC7 protein in all cells in the field of view was significantly decreased.

The sequence of another siPgc7 was added in Supplementary Table 2.

  1. Response b) to Comment 2. Although the authors have readjusted the contrast and definition of the pictures in Figure 4b, it is impossible to conclude specific co-localization of ERK1/2 and DNMT1 in cytoplasm and nucleus because all proteins localize in cytoplasm or nucleus. The authors should remove these figures, change the expressions, or perform additional experiments such as Proximity Ligation Assay to show the interaction between two molecules.

Response 2: Thanks a million for your comments. We have removed the pictures of the localization of ERK1/2 and DNMT1 in Figure 4.

Minor comments:

The authors did not explain “FAM” in the main text.

Response: Thanks a million for your comments. We have explained “FAM” in results section (page 3, line 102) and in the materials and methods (page 12, line 436-437) and marked them with red font.

Round 3

Reviewer 2 Report

The authors have almost satisfactorily modified several points and added some descriptions in the revised manuscript to address the Reviewer’s comments.  This reviewer recommends acceptance of this revised manuscript.

Author Response

   Thanks a million for all comments and suggestions. According to your suggestions, we have refined the language of the manuscript.